# Nonviral Delivery Systems of mRNA Vaccines for Cancer Gene Therapy

**DOI:** 10.3390/pharmaceutics14030512

**Published:** 2022-02-25

**Authors:** Yusi Wang, Rui Zhang, Lin Tang, Li Yang

**Affiliations:** State Key Laboratory of Biotherapy and Cancer Center/Collaborative Innovation Center for Biotherapy, West China Hospital, Sichuan University, Chengdu 610041, China; yusi@stu.scu.edu.cn (Y.W.); ruiruixiaoxiong@foxmail.com (R.Z.); tlin2022@stu.scu.edu.cn (L.T.)

**Keywords:** mRNA, mRNA vaccines, cancer therapy, mRNA design, nonviral delivery system

## Abstract

In recent years, the use of messenger RNA (mRNA) in the fields of gene therapy, immunotherapy, and stem cell biomedicine has received extensive attention. With the development of scientific technology, mRNA applications for tumor treatment have matured. Since the SARS-CoV-2 infection outbreak in 2019, the development of engineered mRNA and mRNA vaccines has accelerated rapidly. mRNA is easy to produce, scalable, modifiable, and not integrated into the host genome, showing tremendous potential for cancer gene therapy and immunotherapy when used in combination with traditional strategies. The core mechanism of mRNA therapy is vehicle-based delivery of in vitro transcribed mRNA (IVT mRNA), which is large, negatively charged, and easily degradable, into the cytoplasm and subsequent expression of the corresponding proteins. However, effectively delivering mRNA into cells and successfully activating the immune response are the keys to the clinical transformation of mRNA therapy. In this review, we focus on nonviral nanodelivery systems of mRNA vaccines used for cancer gene therapy and immunotherapy.

## 1. Introduction

Cancer continues to be a leading global health problem. According to the World Health Organization (WHO), nearly 10 million people died of cancer in 2020, and 7/10 of these people resided in low-income and middle-income countries. It has been estimated that by 2025, nearly 20 million new cancer cases will be diagnosed each year [1].

To date, four major strategies are applied to cancer therapy, including surgery, chemotherapy, radiotherapy, and biotherapy. Biotherapy is an emerging cancer treatment modality that shows significant curative effects, which includes targeted biomolecular drugs and cancer vaccines. The first cancer vaccine was used in 1976, when Morales treated superficial bladder tumors through the vesical and intradermal administration of the Bacillus Calmette–Guerin vaccine, which led to favorable outcomes for nine patients [2]. After decades of effort, the FDA approved the first therapeutic cancer vaccine, Sipuleucel-T (Provenge), which shows efficacy in the treatment of prostate cancer [3]. Currently, the field of cancer vaccines is booming, with a number of potential vaccines under various phases of clinical trials.

Cancer vaccines can be roughly categorized into three types on the basis of the components that trigger the immune system: traditional inactivated tumor cell-based vaccines, synthesized peptide- or protein-based vaccines, and nucleic acid (DNA and mRNA)-based vaccines. Compared with traditional vaccines, nucleic acid-based vaccines have the advantages of a shorter production cycle, easier industrialization, and relatively low prices [4]. However, scientists have discovered that DNA vaccines may integrate into the host genome [5], resulting in insertion mutations, which is the first step towards cancer mutation, ironically. In contrast, mRNA vaccines exert their function in the cytoplasm, which makes them safer [6]. Moreover, the functionality of DNA-based vaccines depends on nuclear envelope breakdown during cell division, while mRNAs can be translated into functional proteins at any point in the cell cycle [7]. In addition, since most viruses have RNA rather than DNA genomes, mRNA is more easy to induce an immune response than DNA [8]. However, this feature of mRNA was once considered to be a trigger for inflammation due to an excessive immune response, but fortunately, scientists developed strategies to balance this immune response. Therefore, mRNA vaccine technology is promising for use in creating a potential cancer vaccine. On the basis of differences between antigen-carrying vectors, cancer vaccines can be classified into four major types: cell-based vaccines (with dendritic cells (DCs) or T cells), virus-based vaccines, bacteria-based vaccines, and molecular vaccines. The first cancer vaccine approved by the FDA was a DC-based vaccine [9]. In this review, we focus on mRNA vaccine functions in cancer therapy, including mRNA delivery into immune cells in vitro with subsequent infusion into patients and direct transfection of mRNA with molecular vectors into target cells.

An examination into mRNA vaccine development history shows that a half century passed from the discovery of mRNA to the clinical application of mRNA vaccines (Figure 1). mRNA was first conceptualized by Jacques Monod and François Jacob and then was demonstrated by Jacob, Sydney Brenner, and Matthew Meselson at California Institute of Technology in 1961. In 1978, Dimitriadis successfully transfected exogenous mRNA into mouse lymphocytes with liposomes [10]. In the same year, Ostro et al. transfected mRNA into human cells [11]. In 1984, Krieg injected biologically active mRNA, synthesized in a laboratory from promoter-containing plasmids, into frog eggs, revealing that the laboratory-made mRNA functioned similar to the endogenous one [12]. Malone et al. first proposed the concept of an mRNA vaccine in 1989 [13], approximately the same time that the cationic lipid-mediated mRNA delivery system lipofectin was commercialized. In 1995, the first cancer immunotherapy study was performed, in which scientists attempted to inject mRNA-encoded cancer antigens into the body to train the immune system to attack cancer cells [14], and this technique was first applied to humans in 2009 [15]. In 2011, transcription activator-like effector nuclease (TALEN) technology was developed for use in gene editing [16], providing powerful tools for mRNA engineering. Moreover, the outbreak of SARS-COV-2 since 2019 accelerated the development of mRNA vaccines. On 11 December 2020, the Pfizer-BioNTech vaccine BNT162b2 received emergency authorization from the FDA and became the first mRNA drug approved for use in humans [17].

Notably, the application of mRNA is limited by instability, innate immunogenicity, and low delivery efficiency in vivo. To overcome these hurdles, appropriate mRNA structural modifications and efficient delivery systems have been studied. Traditional physical systems, such as electroporation (EP) and gene gun, are harmful to cells [18,19]. Viruses are efficacious in delivering mRNA into cells; however, they are not sufficiently safe because they might integrate into the host genome [20]. To balance safety and efficiency, multiple nonviral nanodelivery systems have been developed, including lipid nanoparticles (LNPs), polymers, hybrid NPs, peptides, gold nanoparticle (AuNP)–DNA conjugates, and mRNA-loaded exosomes, all of which show distinct advantages. A comparison of the advantages and disadvantages between viral and nonviral vectors is presented in Table 1.

In this review, we focus on key nonviral nanodelivery systems of mRNA vaccines for cancer gene therapy with an emphasis on the materials, mechanisms, advantages, and limitations of each strategy. Moreover, principles for designing and modifying mRNA to improve its stability are summarized. Finally, we discuss the challenges and future perspectives of mRNA-based cancer vaccines. On the other hand, in this review, we systematically compile the nonviral nanodelivery systems currently used in mRNA vaccines with analysis of their advantages and disadvantages, which may help future mRNA vaccine development on vector selecting. In addition, we also comprehensively list the mRNA cancer vaccines in various clinical trials, providing some updated information.

## 2. Major Types of mRNA Vaccine

mRNA vaccines can be categorized into two main types: self-amplifying mRNA (saRNA) vaccines and nonreplicating mRNA (also called conventional mRNA) vaccines. Both RNA types feature five basic elements of mRNA: a 5ʹ cap, a 5ʹ untranslated region (UTR), an open reading frame (ORF) that encodes an antigen, a 3ʹ UTR, and a poly(A) tail (Figure 2a). In addition to these five elements, a saRNA has a gene sequence encoding the RNA replicase complex (Figure 2b).

### 2.1. Conventional mRNA Vaccine

Conventional mRNA has a relatively small size, with 1000–5000 nucleotides [21], which leads to easier and more effective mRNA encapsulation. Liang et al. demonstrated that a conventional mRNA-based vaccine can effectively induce both innate immunity and adaptive responses [22]. However, nonreplicating mRNA leads to only transient protein expression, and therefore, to achieve effective therapeutic effects, a higher dose of vaccine is required [23].

### 2.2. Self-Amplifying mRNA Vaccine

Compared with conventional mRNA, saRNA is larger, with 9000–12,000 nucleotides, since an additional alphavirus-derived coding sequence is included; this ORF encodes four nonstructural proteins (nsPs), which can converge to form RNA replicases [24], driving the amplification of the mRNA-encoded antigen [25]. Compared with the conventional mRNA vaccine, the saRNA vaccine can induce more effective and durable antigen protein expression, which addresses the transient expression that limits conventional mRNA therapeutics [26]. By comparing the capacities of different vaccines to protect mice against influenza, Vogel et al. demonstrated that a saRNA vaccine shows efficiency equivalent efficiency to that of mRNA vaccines but at a much lower dose, indicating that saRNA is a promising platform for future vaccines [27].

## 3. The Core Mechanism of mRNA Vaccines for Cancer Therapy

The core component of cancer vaccines is an in vitro transcribed mRNA (IVT mRNA), a single-stranded polynucleotide with the same structure and biological activity as endogenous mRNA. The IVT mRNA can encode a tumor-associated antigen (TAA) or tumor-specific antigen (TSA). TAAs are highly expressed in proliferating tumor cells, and normal cells also synthesize small amounts of TAAs. However, vaccines based on TAAs may cause unwanted immune responses in normal tissues [28]. TSAs, also called neoantigens, are expressed only in tumor cells but not in normal cells at any stage [29]. Despite that there are some differences between TAA and TSA mRNA vaccines, the core mechanisms by which they treat cancer are the same.

mRNAs loaded into various vehicles enter antigen-presenting cells (APCs) and follow the conventional endocytic route, trafficking into early endosomes and then to late endosomes, from which they are ultimately recycled, cleared from cells through exocytosis [30], or trafficked into lysosomes, where the mRNA is enzymatically degraded [31]. However, only a small fraction of these mRNA-loaded vectors can escape endosomes and release mRNA in the cytoplasm, which is called endosomal escape, while its specific mechanism is still not clear. When mRNA is released into the cytoplasm, it will induce both the innate immunity and adaptive immunity in two different ways (Figure 3).

On the one hand, mRNAs with pathogen-associated molecular patterns (PAMPs) would be recognized as foreign RNA by specific pattern recognition receptors (PRRs) [32], including Toll-like receptors (TLRs), retinoic acid-inducible gene-I (RIG-I)-like receptors (RLRs), and some newly discovered sensors, such as RNA helicases, heterogeneous nuclear ribonucleoproteins (hnRNPs), and ZBP1, stimulating innate immune response [32]. Endosome-residing TLRs, which identify foreign mRNA before other PRRs, activate some transcription factors; then, these activated transcription factors are translocated into the nucleus to drive the expression of proinflammatory cytokines and type I and III interferons (IFNs). Similarly, RLRs, which are primarily located in the cytoplasm with a small portion located in the nucleus [33], exert the same effects as cytoplasm-residing TLRs [34]. Ultimately, a proinflammatory microenvironment is formed, which induces type 1 helper T cell (TH1-type) immune responses while suppressing TH2-type functions [35].

On the other hand, mRNA can be translated into functional antigen proteins with ribosomes; then it is broken down into small peptide fragments by the proteasome complex or secreted out of the cell. Intracellular peptide fragments are displayed on the cell surface by type I major histocompatibility complex (MHC-I) proteins, which can be recognized by cytotoxic CD8+ T lymphocytes (CTLs). Secreted antigen proteins endocytosed and fragmented by APCs, especially DCs, are presented by MHC class II molecules to CD4+ TH cells. All nucleated cells can potentially process mRNAs and present peptide fragments through MHC-I molecules, but only APCs can present antigens through MHC-I and MHC-II molecules [36].

In addition, both the activated CD4+ TH cells and innate immunity can stimulate the activation of CTLs through the production of inflammatory cytokines; then, abundant CTLs are activated to kill tumor cells, thereby contributing to cancer therapy [37,38].

## 4. The Principles of mRNA Vaccine Design and Modification

mRNA-based vaccines show tremendous potential in the field of cancer therapy. However, because of the easy biodegradability and intrinsic immunogenicity of IVT mRNA, the clinical translation of mRNA vaccines is hindered. Rapid mRNA degradation reduces the efficiency of translation in vivo, leading to a low vaccine titer. The intrinsic immunogenicity has been demonstrated to severely compromise the expression of the desired proteins and mRNA stability by inducing robust type I IFN responses [39] and programmed cell death mediated by substantial overexpression of caspase-1 [40]; however, immunogenicity simultaneously contributes to a positive immune response [41]. To make the mRNA vaccine more efficient, we need to modify the structure and sequence of IVT mRNA to enhance its stability and maintain a moderate immunogenicity.

### 4.1. The 5′-Cap

Discovered in the 1970s, the 5′-cap structure (m7G5′ppp5′N), composed of a 7-methylguanosine nucleoside and a terminal nucleotide linked through a triphosphate bridge in the 5′-mRNA, confers IVT mRNA stability and translation efficiency [42]. There are three major types of 5′-cap structures, including type O, type I, and type II, which are classified according to whether there is a methylated ribose on the 2′ hydroxyl group of the first or second nucleotide from the 5′ end [43] (Figure 4). The 5′-cap structures not only prevent IVT mRNA from being degraded by enzymes, such as alkaline phosphatase (AKP) and 5′ to 3′ exonuclease [44], but also can promote protein biosynthesis by forming starting complexes with the translation initiation factors eIF4E and eIF4G [45,46]. Moreover, the direction of the 5′-cap is very important, as indicated by a previous study demonstrating that mRNA with an inverted cap shows profoundly decreased translation efficiency [47]. Therefore, a 5′-cap oriented in the correct direction is an essential structure in IVT mRNA design. In addition, recent studies have shown that cap modifications can optimize IVT mRNA. For example, Dulmen et al. reported that mRNA carrying a propargyl group at the N6-position of adenosine showed consistent translational efficiency and induced a moderate increase in the immune response [48].

### 4.2. The 5′-UTR

The 5′-UTR is a noncoding region, but it can help mRNA bind to ribosomes [42]. Early research suggested that the secondary structure of the 5′-UTR can inhibit mRNA translation, whose symbol is a high GC content [49,50]. Therefore, IVT mRNAs need to be designed without GC-enriched 5′-UTRs. Moreover, the 5′-UTR should be short and loose to allow small-molecule ribosomes to bind to the initial coding element [43]. In addition, start codons (AUGs) should be avoided in the design of 5′-UTR because they can disrupt ORF translation [51]. Recently, Jia et al. found that adenine nucleotide(A)-rich elements in 5′-UTRs destabilize untranslated mRNAs, although they enable cap-independent translation [52].

### 4.3. The ORF

As the antigen-protein-encoding region, the translation efficiency of ORF is crucial. An early study showed that there is no direct relationship between the length of the ORF and translation efficiency [53]. Some studies demonstrated that GC-rich ORFs showed higher translation efficiency, even though GC-rich sequences may lead to secondary structure formation [54,55]. Codon optimization can also be employed. For example, changing rare codons into common ones in the host without changing the amino acid sequence of the encoded protein can increase the protein expression level [56,57]. In addition, IVT mRNA with modified nucleosides is better, because unmodified single-stranded RNA, the indication of a viral infection, can be recognized by the immune system, resulting in fast mRNA decay. For example, compared with unmodified mRNA, mRNA with pseudouridine (Ψ) produced more proteins by diminishing PKR activation [58,59]. Another study showed that IVT mRNA in which modified nucleosides such as m5C, m6A, m5U, s2U, or pseuduridine were incorporated, induced an attenuated innate immune response, protecting the mRNA from clearance [60]. Recently, Verbeke et al. used nucleoside-modified mRNA (m5C and Ψ) with TLR agonists in cancer therapy, inducing a high degree of T cell immunity without inducing a high level of type I IFN expression [61].

### 4.4. The 3′-UTR

A previous study demonstrated that the length of the 3′-UTR plays an important role in the characteristics of the mRNA, with a longer 3′-UTR exhibiting a shorter half-life and higher efficient translation [62]. In addition, a GU- or AU-rich element can activate rapid IVT mRNA decay [52,63], which is to be avoided. Scientists developed a technology to use alpha-globin and beta-globin 3′-UTRs, which can confer stability on heterologous mRNA in cells [64,65]. Moreover, by performing cellular library screening, Orlandini et al. discovered AES-mtRNR1- and mtRNR1-AES-based 3′-UTRs that can increase the stability of IVT mRNA to enhance the total protein expression, comparable to that of the broadly used human beta-globin 3′-UTR [66].

### 4.5. The 3′-poly(A) Tail

The 3′-poly(A) tail, a sequence absolutely required for mRNA, can increase stability and translation efficiency for mRNA. Early research found that the function of the poly(A) tail may be associated with mRNA breakdown in the cytoplasm [67]. Later, scientists showed that the length of the poly(A) tail is proportional to translation efficiency [68,69,70]. However, a recent study reported that a short tail is a feature of abundant and well-translated mRNAs across eukaryotes [71]. In summary, in an IVT mRNA design, the 3′-poly(A) tail is essential, but the length should vary to endow different mRNAs with an effective translation capacity, which means that there is no universally optimal length of 3′-poly(A) for mRNAs.

## 5. The Nonviral Nanodelivery Systems of mRNA Vaccines

The technique of modifying mRNA can greatly improve its stability and reduce its immunogenicity. Moreover, the delivery systems of the mRNA vaccine are also a key component; ideal vectors need to be capable of effectively delivering mRNA into targeted cells without inducing significant unwanted immune response or toxicity while protecting mRNA from fast degradation. In the following sections, we describe six major nonviral nanosystems used for mRNA delivery (Figure 5).

### 5.1. Lipid Nanoparticles (LNPs)

LNPs are the most clinically advanced mRNA delivery vehicles, and all SARS-CoV-2 mRNA vaccines approved for clinical use are delivered by LNPs [72]. LNPs are composed of a lipid bilayer shell encompassing a hydrophilic core encapsulating the fragile mRNA [73]. Four kinds of lipids are used to form the shell, including cholesterol, helper phospholipids, polyethylene glycol-modified (PEGylated) lipids, and ionizable lipids [74]. Each lipid performs a different function in the NP in mRNA delivery. Cholesterol, a naturally occurring lipid, can enhance the stability of NPs because it fills gaps between lipids [75] and reduce the possibility of immune clearance of the NPs [76,77]. Helper phospholipids are always zwitterionic lipids, such as 1,2-dioleoyl-sn-glycero-3-phosphoethanolamine (DOPE) or 1,2-distearoyl-sn-glycero-3-phosphocholine (DSPC), and sterol lipids [31]; they can enhance the encapsulation efficiency and help LNPs escape endosomes [78]. PEGylated lipids can potentially improve NP manufacturability and stability by forming a hydrophilic layer on the surface of the LNPs to prevent aggregation [79]. Owing to an appropriate pKa, the ionizable lipids can be protonated at an acidic pH to condense the mRNA and subsequently release it inside the cells [80]. The proportion of each kind of lipid is important for forming effective LNPs.

LNPs are prepared by mixing lipids in an ethanol phase and mRNA in an aqueous phase in a microfluidic mixing device [81]. Then, mRNA-loaded LNPs are injected into the body and subsequently enter targeted cells by interacting with negatively charged cell membrane components or specific proteins exposed on the cell membrane [82]. Gilleron et al. demonstrated that LNPs are engulfed by cells through clathrin-mediated endocytosis as well as micropinocytosis [83]. Only a small fraction (1%–2%) of mRNA-loaded LNPs in the cells can evade the endosomal pathway and reach the cytoplasm [83].

Compared with EP, the most common technique used for ex vivo transfection, LNPs are a promising alternative for mRNA delivery because they show low cytotoxicity, stable mRNA cargo, and enhanced intracellular delivery and can be used without the need for specialized equipment [81]. Currently, LNPs are widely used for in vivo delivery of mRNA vaccines. Richner et al. generated an LNP-encapsulated modified mRNA vaccine encoding Zika structural genes to protect against Zika virus infection, and the results showed that it conferred protection and sterilized immunity in immunocompetent mice [84]. An mRNA rabies vaccine (CV7202) formulated with LNPs and studied in a human volunteer clinical phase 1 trial showed that a low dose of the vaccine was well tolerated and induced rabies-neutralizing antibody responses that met WHO criteria in all recipients [85]. Facing the sudden epidemic caused by SARS-CoV-2 infection, scientists developed an mRNA vaccine (BNT162b2) in which LNPs were employed to deliver mRNA [86]. The latest research shows that through 6 months of follow-up and despite a gradual decline in vaccine efficacy, BNT162b2 showed a favorable safety profile and was highly efficacious (the efficacy against SARS-CoV-2 infection was 91.3%) in preventing the development of COVID-19 [87]. To date, many mRNA vaccines employing LNPs as mRNA carrier vehicles have undergone clinical evaluation for cancer therapy.

However, some hurdles have limited the application of LNPs. One major hurdle is the safety associated with LNP formulations. Having evaluated the safety of mRNA-loaded LNPs in a Sprague–Dawley rat and cynomolgus monkeys, Sedic et al. concluded that the toxicological effects and possibly the distribution properties of the drug product are predominantly vehicle driven [88]. Moreover, the lipid components of the LNPs may activate host immune responses following systemic or local administration. However, the immunogenicity of the lipids can be regarded as a double-edged sword. On the one hand, it acts as a self-adjuvant, promoting antigen presentation in patients who receive mRNA vaccines [89]. On the other hand, it may contribute to severe side effects. For example, Kelso reported that the PEG in the LNPs of the Pfizer-BioNTech mRNA COVID-19 vaccine might cause anaphylaxis in patients with IgE-triggered allergies [90]. Moreover, PEG would inhibit the fusion between LNPs and cell membranes [91], decreasing mRNA uptake efficiency. Cationic and ionizable lipids have also been reported to stimulate the secretion of proinflammatory cytokines and reactive oxygen species [92].

Despite some insufficiencies, mRNA-loaded NPs are still potential clinical tools, as demonstrated by the unprecedented rapid development of mRNA COVID-19 vaccines. To make LNPs more powerful in mRNA delivery, many scientists tried to optimize them, and there have been some new findings. For transfection efficiency, Kauffman et al. showed that increasing the ionizable lipid–mRNA weight ratio can enhance delivery efficiency [93]. Ball et al. optimized the LNP delivery system by adding a negatively charged “helper molecular” to the NP. In their study, the mRNA-loaded LNPs coformulated with siRNA induced three-fold increases in luciferase protein expression compared with that in formulations without siRNA [94]. For endosomal escape, Herrera et al. found that it preferentially occurs in late endosomes, not early endosomes [95]. Maugeri et al., demonstrated that endosomal escape of mRNA-loaded LNPs depends on the molar ratio between the ionizable lipids and mRNA nucleotides [96]. Lee et al. studied the interaction between LNPs and a model endosomal membrane and showed that 4A3-Cit (a lipid with an unsaturated tail) exhibited superior lipid fusion over saturated lipids, suggesting that unsaturated lipids promote endosomal escape [97]. For targeted delivery, scientists have observed that the targeting functionalities of LNPs are largely related to the chemical structure of the active lipids [98]. For instance, imidazole-based LNPs preferentially target splenic T cells [99]; Zukancic et al. found that PEGylation is critical for achieving selective organ targeting, even though it is at the lowest ratio in LNPs [100]. For immunogenicity, Hassett et al. found that the particle size may influence LNP immunogenicity; they demonstrated that LNPs at a smaller diameter were substantially less immunogenic in mice, but all the particle sizes tested induced a robust immune response in nonhuman primates (NHPs), suggesting that an optimal mRNA vaccine particle size as determined for rodents may not translate to primates [101]. For toxicity, previous research has shown that nonbiodegradable lipids would cause mortality in mice, whereas biodegradable and nonbiodegradable lipids administered at a similar dose were well tolerated [102].

LNPs show great potential in mRNA delivery and are expected to become the preferred delivery carrier of mRNA vaccines. It is also expected that the development of new LNPs will improve mRNA vaccine safety and efficacy to protect human health.

### 5.2. Polymer Nanoparticles

Polymer NPs were among the earliest carriers used for nucleic acid delivery [103]. Since the first successful delivery of foreign DNA to liver cells with a cationic polymer carrier system [104], polymer NPs have been rapidly developed and widely applied. There are three major kinds of polymer NPs: cationic polymer, dendritic, and polysaccharide NPs.

Cationic polymers are easy to generate and flexible to modify, attracting considerable attention in the field of mRNA delivery. Cationic polymers and negatively charged mRNA self-assemble under aqueous conditions by ionic and hydrogen bonding to a complex. It has been shown that increasing the molecular weight of the cationic polymer enhances the efficiency of saRNA delivery both in vitro and in vivo [105]. The most characterized cationic polymer is polyethyleneimine (PEI). PEI is a highly branched network polymer (25% consists of key nitrogen atoms) that can capture mRNA with high efficiency [106]. It contains low pKa amines, which can help on endosomal escape through protonation in the endosomal acidic compartments, making PEI the gold standard cationic polymer used for endosomal escape of nucleic acids [107]. In addition, investigations have revealed that PEI with a high molecular weight binds mRNA too tightly to induce protein expression, while PEI with a low molecular weight (2 kDa) shows higher protein expression [108]. Nevertheless, early studies have shown that PEI is potentially cytotoxic, inducing rapid plasma membrane disruption that resembles early necrotic-like changes and activates a “mitochondrially mediated” apoptotic program in later stages [109,110]; therefore, PEI did not enter the clinical phase of testing [111]. To improve the safety profile of this carrier, scientists have tried to modify PEI. Zhao et al. reported a stearic acid and branched PEI-2k conjugate (PSA), showing effective mRNA delivery and antigen-specific immune response; more importantly, PSA was less toxic than PEI-25k [112]. Li designed a cationic cyclodextrin–PEI 2k conjugate (CP 2k) to deliver mRNA encoding HIV gp120. The linking cyclodextrin to PEI lowered the charge density of the polyamine backbone and thus reduced the cytotoxicity while maintaining many protonatable groups, which led to high delivery efficiency [113].

Dendritic polymers are flexible macromolecules with a diameter ranging from 2 to 20 nm. They are composed of a core and multiple regularly hyperbranched monomers possessing a large number of functional groups that can be modified to influence the properties of the polymer (e.g., solubility, tissue binding, and pharmacokinetics) and to carry molecules via labile chemical linkages. As the polymer weight is increased, the terminal units are packed more closely, creating effective drug payloads [114,115]. PAMAM dendritic polymers, which were the first to be synthesized, can be modified for use in delivering nucleic acids [116]. However, there are some drawbacks of the dendritic polymers: rapid systemic elimination; inefficient accumulation in the targeted organ [117]; and significant toxicity, including hemolytic toxicity, cytotoxicity, and hematological toxicity [118]. To overcome these barriers, Zhang et al. developed an ionizable amphiphilic Janus dendrimer (IAJD) delivery system, which exhibited high activity with a low concentration of ionizable amines, and it seems to possess the highest mRNA delivery efficiency into the lungs to date [119].

Polysaccharides are relatively common natural biomaterials that can be produced at low cost and high biocompatibility and biodegradability with little obvious cytotoxicity in vivo. Similar to dendritic polymers, polysaccharides can be easily chemically modified for efficient delivery [120]. However, polysaccharide polymers show some weaknesses. For example, since polymers are natural materials with variable molecular weight and components, the structure is difficult to delineate. Additionally, they lack solubility in most solvents [121]. However, these drawbacks did not hinder the development of polysaccharide particles for use in mRNA vaccines. Many groups have studied the capacity of chitosan-based NPs to deliver mRNA, and they have demonstrated that these NPs are very effective and safe, while inducing no apparent cytotoxicity [122,123,124,125], showing enormous potential for use in nanomedicine.

These three kinds of polymer NPs are not completely used independently; sometimes they are used in combination to get a higher efficiency and adequate safety. In addition, some polymers cannot be simply classified into any category. Sharifnia et al. developed a PLGA/PEI nanoparticle, whose encapsulation efficiency exceeded 73.5%; moreover, it provided effective delivery of IVT mRNA encoding GFP without cytotoxicity detected [126]. Naoto Yoshinaga et al. designed a system composed of only PEG and mRNA, and it showed enhanced resistance against RNases and effective reporter protein expression in cultured cells, which is nearly 20-fold higher than naked mRNA [127].

### 5.3. Polypeptide Nanoparticles

Polypeptides are macromolecules polymerized by amino acids. They are highly biodegradable and exhibit high biocompatibility and low cytotoxicity.

Protamine, a family of small peptides with a molecular weight of 4000 Da, can be obtained from fish sperm. It is positively charged, showing excellent mRNA-binding capability through electrostatic interactions of its arginine-rich sequences [128]. The application of protamine is very early, which was demonstrated to be effective in delivering RNA 50 years ago [129]; moreover, it is relatively mature, widely used in the developing mRNA vaccine. For instance, CV9201, an mRNA-based vaccine that employs protamine as the vehicle, showed acceptable tolerability and moderate immune activation for non-small-cell lung cancer (NSCLC) in clinical trials [130].

In addition to protamine, cell-penetrating peptides (CPPs), which were identified 20 years ago, are potential mRNA vehicles [131]. CPPs are cationic with a relatively short sequence of amino acids (less than 30 residues) [132]. Most CPPs are derived from natural proteins of different origins, while they all have a similar ability to transport different molecules across biological cell membranes with low cytotoxicity [133]. PF14, an amphipathic CPP consisting of 21 amino acids with N-terminal stearylation, achieves efficient mRNA expression with relatively low concentrations. Moreover, PF14 can induce detectable reporter protein expression in a xenograft model of ovarian cancer, which is impossible for standard commercially available LNPs [134]. Udhayakumar et al. developed a CPP containing the amphipathic RALA motif, showing a relatively high expression of antigen proteins in DC cytosol and efficient induction of potent CTL responses compared with those induced by a standard liposomal mRNA formulation (DOTAP and DOPE) [135]. Protein is a special kind of polypeptide, so it can fit into this category. Segel et al. recently designed a new mRNA delivery platform, selective endogenous encapsidation for cellular delivery (SEND), in which the core is the mammalian retrovirus-like protein PEG10. PEG10 is a naturally occurring protein with low immunogenicity, which can bind to its own mRNA, forming a spherical capsule. By flanking genes of interest with PEG10′s untranslated regions, the mRNA cargo of PEG10 can be reprogrammed to realize mRNA delivery. The results showed that SEND is a modular platform for efficient mRNA delivery, and it allows repeated administration [136].

Although polypeptide NPs have been rapidly developed in recent years, this delivery system has not been applied in the clinic. However, they show great prospects in the field of biomedicine.

### 5.4. Hybrid Nanoparticles

Hybrid nanoparticles are composed of various materials, such as lipids, polymers, and peptides, converging the advantages of each component [137]. Compared with the counterpart monosystem, these hybrid NPs are more stable and efficient in mRNA delivery [138].

Commonly used hybrid NPs include lipid–polymer hybrid nanoparticles (LPNs), peptide–lipid hybrid NPs, peptide-polymer hybrid NPs, and peptide-polymer-lipid hybrid NPs. LPNs are the earliest reported example of hybrid NPs. Polymer NPs exhibit high structural integrity, stability, targeted delivery, and controlled release capability; however, they also inevitably exhibit cytotoxicity [139]. LNPs are highly biocompatible with the drawback of ease of clearance by the reticuloendothelial system. To compensate for the limitations of both lipid and polymer NPs, the two types of NPs were combined to generate the LPN [140]. Siewert et al. assembled an LPN using the cationic lipid, DOTAP, and protamine, and they achieved highly efficient mRNA transfection. In addition, they found that, compared with sequentially assembled particles, delivery systems prepared by mixing the complexing agents before adding mRNA resulted in hybrid NPs with an inhomogeneous internal organization, showing greater transfection efficiency [141]. Many studies have reached similar conclusions, further demonstrating that LPNs are superior to single-system NPs [142,143,144]. For the use of peptide-lipid hybrid NPs, Zhang et al. designed a peptide-lipid hybrid NP composed of DOTAP and the cholesterol-modified cationic peptide DP7 (VQWRIRVAVIRK), improving the efficiency of intracellular mRNA delivery and increasing the immune response, thus enhancing the antitumor effect of neoantigen mRNA vaccines [145]. In the application instance of peptide-polymer hybrid NPs, Coolen et al. constructed PLA-CPP NPs, in which the CPPs were added to adsorb negatively charged mRNA onto the negatively charged PLA NP surface. PLA is a biodegradable and biocompatible polymer with a high safety profile and efficient internalization by DCs in vitro and in vivo. Comparing three different CPPs (RALA, LAH4, and LAH4-L1) to serve as cationic intermediates, they found that LAH4-L1 and PLANP/LAH4-L1 induced the highest protein expression in DCs [146]. In addition, Qiu et al. introduced a novel mRNA vector, PEG12KL4, composed of a monodisperse linear PEG consisting of 12-mers attaching a synthetic cationic KL4 peptide. After intratracheal administration, the analysis showed that it is superior to naked mRNA or Lipo2k, not only in transfection efficiency but also in targetability [147]. For the example of peptide-polymer-lipid hybrid NPs, by fusing cRGD-R9 (a CPP) and a DMP gene vector backbone synthesized from a combination of DOTAP lipids and mPEG–PCL polymers, Gao et al. developed an advanced mRNA delivery system, DMP-039. By investigating the in vivo distribution, degradation, and excretion of the mBim/DMP-039 complex in detail, they concluded that the synthesized DMP-039 hybrid NPs possess high mRNA delivery capacity and good safety profile while showing effective suppression of pulmonary metastatic tumor progression [148].

In addition, there are some rare kinds of hybrid NPs. For instance, Choi et al. reported on a graphene oxide (GO)-PEI hybrid NP (an inorganic-metal-organic hybrid particle) that can deliver mRNA effectively, leading to efficient generation of “footprint-free”-induced pluripotent stem cells (iPSCs) without the potential risk of insertional mutagenesis in humans [149].

### 5.5. Gold Nanoparticle–DNA Oligonucleotide Conjugates

Gold nanoparticle–DNA oligonucleotide conjugates (AuNP–DNA conjugates) are composed of an individual AuNP and several DNA oligonucleotide strands that are modified to contain a thiol or disulfide group that attaches to the Au surface of the conjugates [150]. AuNP is a solid ball with small diameters, from 5 to 100 nm, that are made through the reduction of chloroauric acid [151]. Moreover, AuNP is highly biocompatible with low cytotoxicity, which is a result of using an inert Au core [152]. In addition, AuNP has a very large surface-to-volume ratio, and these large surface areas can be modified by additional diverse biomacromolecules, which can reduce dose-dependent side effects [153]. However, in tissues where the rate of clearance from circulation is relatively low, AuNPs might cause health problems, indicating that specific targeting must be achieved [151]. In 1996, Alivisatos et al. first developed AuNP–DNA conjugates by attaching several DNA strands of a defined length and sequence to the surface of individual AuNPs [154]. AuNP–DNA conjugates are generally formed by covalent binding, and there are many studies that have described alternative protocols for the preparation of them with discrete, well-defined, homogeneous, and soluble physicochemical properties [155,156,157]. The DNA sequences attached to the AuNPs guide the conjugates to targeted cells, where they combine with their complementary sequences following the Watson–Crick principle, thus achieving precise delivery. For example, Ye et al. designed AuNP–DNA conjugates in which the DNA sequences complementarily hybridized with survivin, an mRNA related to apoptosis inhibition that is overexpressed in most cancer cells but undetectable in normal cells, leading to an increased cancer cell death rate [158].

The AuNP–DNA conjugates are rarely used for mRNA vaccines even though they have shown considerable potential in the field, but some progress has been made. Notably, Yeom et al. engineered BAX mRNA-loaded AuNP–DNA conjugates and then injected them into xenograft tumors in mice, resulting in highly efficient mRNA delivery and biologically functional BAX protein (a proapoptotic factor) production, subsequently inhibiting tumor growth [159]. Moreover, Zhang et al. demonstrated that DNA-attached AuNPs are more easily taken up by cells and are highly stable in serum-containing solutions [160].

### 5.6. mRNA-Loaded Exosomes

Exosomes are membrane structures surrounded by a lipid bilayer with a relatively small diameter, from 40 to 400 nm [161]. Exosomes were first described in 1981. Trams, E. G., et al. found some vesicles with 5′-nucleotidase activity in cultures that had been shed from various normal and neoplastic cell lines. In these shed microvesicles, the amounts of sphingomyelin and total polyunsaturated fatty acids in phospholipids were significantly increased. Hence, they concluded that the shed vesicles contained a portion of the plasma membrane [162]. Since exosomes are sprouted from the cell membrane with a quite small diameter, not polymers synthesized by cells, they are highly biocompatible without inducing adverse effects in vitro or in vivo at any dose tested, and they can overcome biological barriers that normal vectors cannot pass, such as the blood brain–barrier (BBB) [163]. By evaluating the potential cytotoxic effects of HEK293T-derived exosomes on the THP-1 and U937 human monocytic cell lines, Rosas, L. E., et al. confirmed that exosomes are safe for use in the clinic [164]. Isolated exosomes can be transmitted everywhere in the body, and targeted distribution can only be realized when specific surface-derived targeting molecules from parent cells are on the exosomes [165]. In addition, exosomes can evade the immune system, prolonging their circulation time in the body, which provides sufficient time for exosome-loaded molecules to exert their functions [166].

In the field of mRNA vaccines, exosomes have natural advantages for efficient delivery, leading to abundant research on mRNA-loaded exosomes. For instance, using exosomes obtained from HEK293 cells, an immortalized human cell line, Tsai et al. transfected mRNA encoding red-light-emitting luciferase Antares2 into human cells, and the exosomes showed higher transfection efficiency than LNPs. Moreover, the research group reported that repeated injection of Antares2 mRNA-loaded exosomes drove sustained luciferase expression without signal attenuation or adverse responses [167]. Forterre et al. designed HchrR6 mRNA-loaded exosomes with considerable targeting, because there are HER2 receptor targeted peptide ligands, EVHB, on the surface of the exosomes. HChrR6 mRNA encodes an enzyme that converts CNOB into the cytotoxic drug MCHB to treat HER2+ human breast cancer. The results of in vivo experiments with mice showed that the growth of orthotopic BT474 xenografts had been nearly arrested [168].

However, large-scale exosome production for clinic applications is difficult and expensive [169]. To boost exosome production, Kojima et al. screened HEK-293T cells and performed in vivo experiments, and they thus identified three genes, STEAP3 (involved in exosome biogenesis), syndecan-4 (SDC4; supports budding of endosomal membranes to form multivesicular bodies), and a fragment of L-aspartate oxidase (NadB; which possibly boosts cellular metabolism by tuning the citric acid cycle), as potential synthetic exosome production boosters [170]. In addition, Yang et al. reported a cellular nanoporation method (CNP) that enables simultaneous delivery of plasmid DNA into source cells for the production of large quantities of exosomes containing therapeutic mRNAs and targeting peptides [171]. Compared with siRNA and microRNA (miRNA), the packing and delivery efficiency of mRNA with exosomes are not enough. Therefore, Kojima et al. developed an active mRNA-packaging device (the archaeal ribosomal protein L7Ae) and a delivery helper device (a gap junction protein, connexin 43) to enhance the efficiency of mRNA transfection into target cells. Consistent with their design, after cotransfecting these two potential devices with the exosome production booster and mRNA-loaded exosomes into the cells, strong luminescence was detected, showing that the functionality of each device was indispensable [170]. Moreover, there is a risk of horizontal gene transfer when exosomes derived from immortalized cells, which might contain some dangerous genes, such as oncogenic DNA and retrotransposon elements, are used [172]. Usman et al. suggested generating exosomes with mature human red blood cells without nuclei, which would completely abrogate the risk of gene transfer. This strategy is practicable, as they successfully generated large-scale amounts of RBC-derived exosomes with effective mRNA delivery [173]. More detailed information on some of the mRNA delivery systems mentioned in this section is listed in Table 2.

## 6. Clinical mRNA Vaccines for Cancer Therapy

mRNA vaccine studies have been performed for decades with the aim of developing cancer therapy. With the development of nanotechnology, the mRNA vaccine field is maturing. Currently, it is incorporated into mainstream research directions for cancer gene therapy. A large number of mRNA cancer vaccines have been completed or are in clinical trials. To date, two basic kinds of mRNA cancer vaccines are being studied for clinical application. In one type, mRNA is transfected into DCs in vitro and reinfuses the mRNA-loaded DCs into the body; in the other type, mRNA in a delivery system is directly injected in vivo. Both strategies show feasibility and tremendous potential for use with mRNA vaccines applied to cancer therapy, as demonstrated in many studies. In this section, we most focus on the clinical and preclinical trials of these mRNA cancer vaccines, which can be found on the website at https://clinicaltrials.gov.

### 6.1. mRNA Cancer Vaccines Based on the Transfection of DCs In Vitro

Dendritic cells (DCs) are the most efficient APCs and are powerful tools used for stimulating the immune system since they can easily capture, process, and present antigens to T cells, thus readily eliciting TH and killer T cells [174]. For DC-based mRNA cancer vaccines, DCs are extracted from the patient’s peripheral blood. Then, granulocyte-macrophage colony-stimulating factor (GM-CSF) and interleukin-4 (IL-4) are used to stimulate the differentiation and maturation of the DCs. Next, IVT mRNAs encoding tumor antigens are transfected by electric pulse or other delivery approaches in vitro. Finally, the mRNA-loaded DCs are reinfused into the patient, thus stimulating the immune system to attack cancer cells [175]. To date, the most DC-based mRNA cancer vaccine transfected the antigen-encoding IVT mRNA into DCs by EP, although it has been demonstrated that using delivery systems, such as LNPs and polymers, can greatly increase the transfection efficiency of naked mRNA [176].

An early study demonstrated that this strategy of DC-based mRNA vaccines is an effective and safe way to induce CTLs and tumor immunity, thus expanding the potential application of DC-based mRNA vaccines to patients bearing small tumors [177]. Moreover, Zhang et al. demonstrated that neoantigen-pulsed DC vaccines were superior to neoantigen-adjuvant vaccines in both activating immune responses and inhibiting tumor growth with the same antigen [178], indicating that DC-based mRNA vaccines show obvious therapeutic advantages over direct injection. There are many completed or active clinical trials with positive results, further proving the effectiveness of mRNA vaccines for cancer therapy. For example, Batich et al. recently used a pp65 mRNA-pulsed DC vaccine admixed with GM-CSF to treat 11 patients with newly diagnosed glioblastoma after they received dose-intensified temozolomide (DI-TMZ). The results of this clinical trial (NCT00639639) showed that patients receiving the pp65-mRNA-loaded DC vaccine had long-term progression-free survival (25.3 months) and overall survival (41.1 months), although the regulatory T cell (Treg) proportions were increased following DI-TMZ [179]. Moreover, Wang et al. developed a DC vaccine against personalized TAAs to treat patients with glioblastoma multiforme (GBM) or advanced lung cancer in combination with low-dose cyclophosphamide, polyinosinic–polycytidylic acid (poly I:C), imiquimod, and an anti-PD-1 antibody (NCT02808364). A total of 10 patients received the treatment, and 7 showed anti-TAA T cell responses without grade III/IV adverse events, and their overall survival was more favorable than that of patients who received standard treatment at the same institution [180]. Vik-Mo et al. conducted a phase I/II clinical trial (NCT00846456), in which they used a DC vaccine with mRNA from tumor stem cells to treat glioblastoma and brain tumor. In 7 of the 20 enrolled participants, an immune response without adverse autoimmune events or other side effects was successfully induced; in addition, compared with matched controls, progression-free survival was 2.9 times longer [181]. All these positive clinical trials show the high potential of DC-based mRNA vaccine in the field of cancer therapy.

However, clinical research studies about it are not absolutely smooth sailing for various reasons. On the one hand, production of DC-based vaccines is expensive and complex; on the other hand, personalized treatment schemes are always needed for different patients, which is quite difficult to get fruition in a short time. Additionally, there are some clinical trials that have been terminated. For instance, the trials numbered NCT00929019 and NCT00514189 were terminated because of slow accrual; as for the reason of logistical problems, the trial numbered NCT00961844 was also terminated. These difficulties cannot hinder the development of DC-based mRNA vaccine since there are still multiple associated clinical trials under active and recruiting stage. For its future development, these complications need to be resolved.

More information about clinical trials performed to test mRNA vaccines mediated by DCs are listed in Table 3.

### 6.2. mRNA Cancer Vaccine Based on Direct Injection In Vivo

After the in vivo injection of mRNA cancer vaccines with nonviral delivery systems, a small fraction of the vehicle-mRNA formulations is taken up by APCs. The few mRNAs that escape endosomes induce the immune response. Compared with the DC-based mRNA cancer vaccine, the transfection efficiency of direct injection with nonviral delivery systems is lower, but this approach is much more convenient, making large-scale production both possible and more affordable. In this strategy, LNP is the most mature tool, and also the most used one; likewise, peptide-based vector is a competitive candidate.

There have been some positive results. In a phase I/IIa study (NCT00923312) of the mRNA-based cancer vaccine CV9201 for non-small-cell lung cancer (NSCLC), 46 patients with locally advanced (7 patients) or metastatic (39 patients) NSCLC and at least stable disease received five intradermal CV9201 injections. The results of the phase IIa trial showed that the 2- and 3-year survival rates were 26.7% and 20.7%, respectively. CV9201 was well tolerated and effectively induced immune responses, supporting further clinical investigation [130]. For the melanoma FixVac (BNT111), an intravenously administered liposomal RNA (encoding antigens: NY-ESO-1, tyrosinase, MAGE-A3, and TPTE) vaccine developed by BioNTech, the data from an exploratory interim analysis showed that melanoma FixVac, alone or in combination with blockade of the checkpoint inhibitor PD1, mediated durable objective responses (NCT02410733) [182]. Moreover, in a phase I trial of Vvax001 (NCT03141463), a saRNA encoding HPV-derived tumor antigen cancer vaccine against human papillomavirus (HPV)-induced cancers, immunological activity, safety, and tolerability were detected. Among the 12 participants with a history of cervical intraepithelial neoplasia, all showed a positive vaccine-induced immune response after immunization, indicating that Vvax001 is safe and effective as a therapeutic vaccine for use in HPV-related malignancies [183]. However, compared with the DC-based mRNA cancer vaccines, the number of clinical trials of direct-injection-based vaccines is fewer, and most of them are still under the recruiting stage. Additional information of clinical trials of this kind of vaccines is listed in Table 4.

## 7. Conclusions and Future Directions

Because of their unique advantages, mRNA vaccines have attracted extensive attention in the field of cancer therapy. However, no mRNA vaccine has been approved by the FDA on the market before 2020. The first mRNA vaccine, BNT162b2 against SARS-CoV-2, received emergency authorization from the FDA on 11 December 2020, after only 8 months of research, breaking the record for the shortest time needed for vaccine development [17]. The success of BNT162b2 is creating an environment to rapidly expand the application of mRNA vaccines to cancer therapy. Currently, the three giants in the field of mRNA vaccines, Moderna, BioNTech, and CureVac, are quite interested in developing mRNA cancer vaccines. To date, an mRNA vaccine for internal melanoma, neuroendocrine tumor, and neuroepithelioma uveal diseases, DCaT–RNA, entered phase III clinical trials (NCT01983748), the last step before approval, with 200 patients. This vaccine is based on autologous tumor RNA-loaded autologous DCs, and the trial is expected to be completed in 2023.

Although some progress in mRNA vaccine development has been made in the field of cancer therapy, research is still in the early stage, multiple insufficiencies hindering the development of mRNA cancer vaccines. First, there is room for improvement of existing delivery systems employed for mRNA vaccines in cancer therapy. To date, the most commonly used vector is LNP, a quite mature tool [184]. However, Ndeupen et al. recently found that the lipid NP component in the mRNA-LNP platform induced high levels of inflammation in mice, regardless of whether the administered LNPs were delivered intravenously, intramuscularly, or intranasally, while the mechanism was not clear [185]. Additionally, it can cause severe allergic reactions [90]. A previous study demonstrated that cationic lipids in LNPs might react with negatively charged biomacromolecules in vivo, inducing severe side effect [92]. Later, scientists developed ionizable LNPs to ameliorate this side effect. However, a recent study showed that these ionizable lipids may include impurities, leading to a loss of mRNA activity, while these impurities are difficult to identify with the traditional techniques [186]. In addition, due to some shortcomings of the LNPs, such as easy oxidation and degradation, nonuniform preparation protocols, and high recurrence rate, large-scale industrial production has been difficult to achieve [187]. Of course, there are many other NPs, but none has been approved by the FDA for clinical use for various reasons. Ideal delivery systems should exhibit high transfection efficiency, sufficient safety, protection of mRNA against fast degradation, and targeted delivery; these features are far from being achieved. Second, the currently used administration routes limit the delivery efficiency of mRNA-vehicle complexes. mRNA cancer vaccines can be administered systematically or locally, and different administration routes can affect the efficiency of target antigen protein translation, leading to different degrees of immune response [188]. The common systematic routes of delivery are intravenous (i.v.), intramuscular (i.m.), hypodermic (i.h.), and intradermal (i.d.) injection. However, mRNA-NP platforms delivered by i.v. can be trapped mainly in the liver, with only a few complexes, reaching target tissues [189]. Administration by i.h. or i.d. does not typically deliver a large therapeutic dose, which can only be achieved through point injection or application of multiple doses. Intramuscular injection (i.m.) accommodates large dose administration, but the requirements for the size and zeta of the mRNA-loaded particles are strict because a large particle size and charge reduce vaccine efficiency. The common local injection routes for mRNA-loaded nanoparticle delivery include intraperitoneal injection (i.p.), which has been shown to be effective in colon cancer therapy [190]. Notably, the local injection strategy has little effect on metastatic cancer. Through currently used administration routes, few mRNA molecules successfully enter the cytoplasm to express proteins, and high-dose administration, with accompanying severe side effects, is still the norm, and a new protocol with an optimal route for mRNA cancer administration is needed. Last but not least, problem with patents has enabled some companies in the field of mRNA vaccines, such as Moderna, to stay in front of the storm, indicating that adequate attention should be directed to problems with scientific research patents. Once the obstacles are overcome, mRNA vaccines will enter a new age and be powerful tools in cancer therapy.

In our opinion, the future directions of mRNA cancer vaccine are personalized vaccines and incorporation with traditional treatment or antitumor drugs. Many mRNA encoding TAAs involve lack of specificity, which may attack on normal cells, yielding disappointing results [191]. Therefore, the employment of specific tumor antigens is necessary for vaccines developing in the future. Owing to the development of sequencing technology and prediction algorithms, finding TSAs is quite fast and easy [192]. However, many TSAs are unique to individuals, indicating that the design of IVT mRNA needs to be individualized to make mRNA vaccines with the maximum antitumor efficiency. Therefore, individualization is the direction for the development of therapeutic mRNA cancer vaccines [193]. Tumor cells can avoid clearance by the immune system through a series of mechanisms, such as developing an immunosuppressive microenvironment and expressing programmed cell death ligand-1 (PDL-1) on the cell surface to counteract T cells. To reverse this immunosuppression, mRNA cancer vaccines are always used in combination with drugs to boost the immune response, such as immune checkpoint inhibitors (ipilimumab, anti-CTLA-4, pembrolizumab, and anti-PD-1) [194]. For example, a clinical trial (NCT03313778) is ongoing, in which the mRNA vaccine was administered in combination with pembrolizumab to treat both unresectable solid tumors and resected cutaneous melanoma. Preliminary results showed acceptable safety and obvious specific T cell responses. In addition, mRNA vaccines can be combined with traditional treatment methods, such as surgery, chemotherapy, and radiotherapy, thereby effectively activating CTLs to attack target tumors [195]. Combination therapy is trending in the application of therapeutic mRNA cancer vaccines.

In summary, mRNA vaccines play important roles in cancer therapy and show huge promise for continuously improving cancer treatment.

## Figures and Tables

**Figure 1 pharmaceutics-14-00512-f001:**
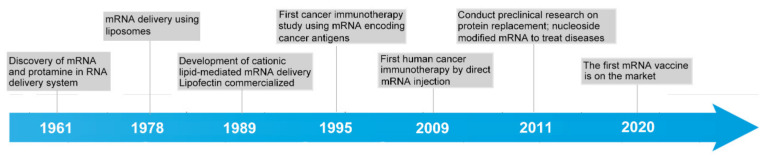
The history of the development of mRNA vaccine.

**Figure 2 pharmaceutics-14-00512-f002:**
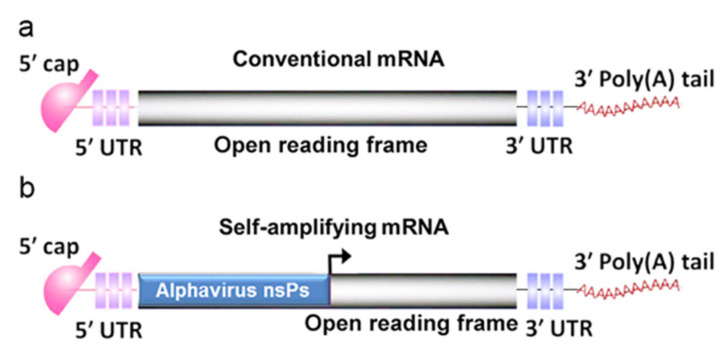
Key structures of in vitro transcribed mRNA (IVT mRNA). (**a**) Conventional mRNA is composed of a 5′ cap, a 5′ untranslated region (UTR), an open reading frame (ORF), a 3′ UTR, and a 3′ poly(A) tail. (**b**) Self-amplifying mRNA includes a 5′ cap, a 5′ UTR, an ORF, a 3′ UTR, a 3′ poly(A) tail, and an additional alphavirus sequence encoding nonstructural proteins (nsPs).

**Figure 3 pharmaceutics-14-00512-f003:**
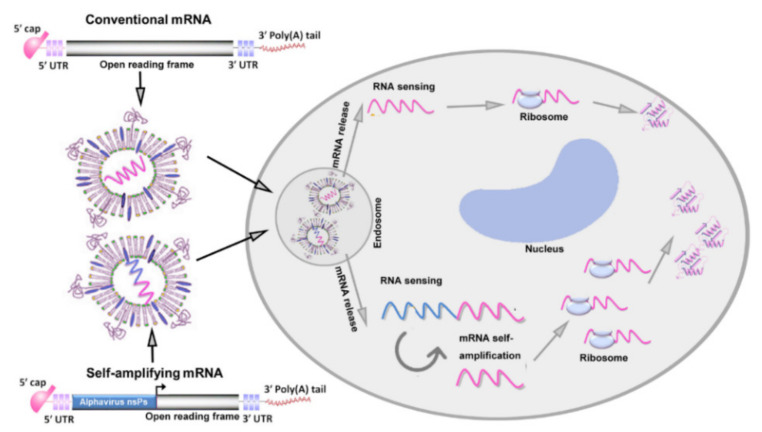
Mechanism of mRNA vaccine cancer therapy. Both conventional mRNAs and self-amplifying mRNAs (saRNAs) encoding antigen proteins are encapsulated in NPs and delivered into cells through the cell membrane. Then, they are trafficked into the endosomes. Only a small fraction of these mRNA-containing NPs escape endosomes and are released into the cytoplasm. Conventional mRNA is sensed by the cell and translated through ribosomes into antigen proteins that can induce an immune response. saRNA undergoes self-amplification that is facilitated by nonstructural proteins (nsPs), leading to the translation of more mRNAs.

**Figure 4 pharmaceutics-14-00512-f004:**
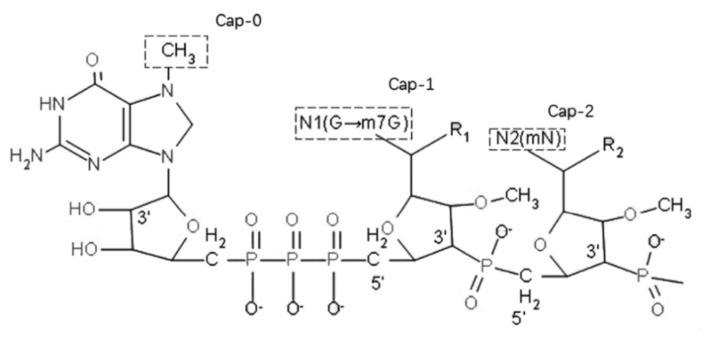
The structure of the 5′-cap (cited from [43]). The first nucleoside in the 5′-cap is usually composed of a guanine methylated at the seventh position and a ribosome (m7G). m7G is connected to the terminal nucleotide of the mRNA through a phosphate bond. The following first or second nucleotide can also be methylated at the 2′ hydroxyl group of the ribose. Type O (m7G5′ppp5′Np) has an unmethylated ribose; type I (m7G5′ppp5′NmpNp) has a methylated ribose in the first nucleotide at the terminus; and in type II (m7G5′ppp5′NmpNmp), both nucleotides are methylated.

**Figure 5 pharmaceutics-14-00512-f005:**
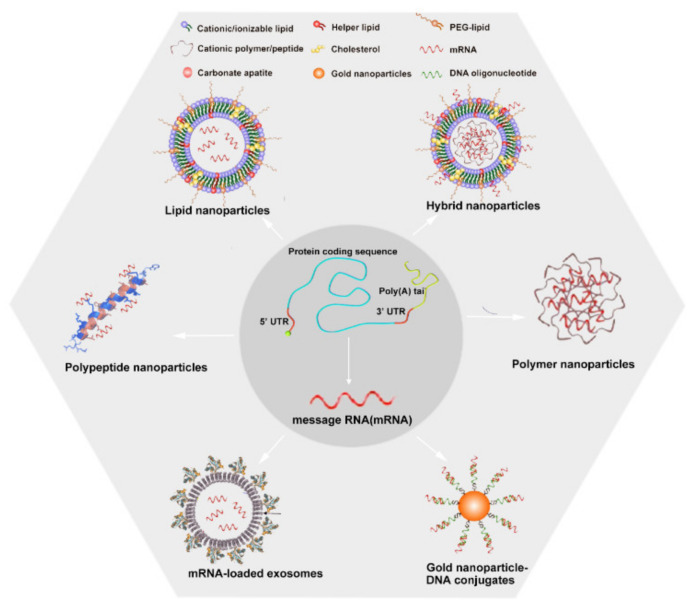
Nanosystem approaches for mRNA delivery. There are six major kinds of nanoparticles (NPs) used as delivery systems for mRNA cancer vaccines: including lipid nanoparticles (LNPs), hybrid NPs, polymer NPs, gold NP–DNA oligonucleotide conjugates (AuNP–DNA conjugates), mRNA-loaded exosomes, and polypeptide NPs. In these systems, the mRNA has the same structure as endogenous eukaryotic mRNA with the five basic elements described herein.

**Table 1 pharmaceutics-14-00512-t001:** Advantages and disadvantages of viral and nonviral vectors.

Vectors	Advantages	Disadvantages
Viral	High transfection efficiency, sustained expression of vector, the complex assembly process is completed by the cells	Elicit immune response, risk of insertion, carcinogenesis, broad tropism, limited DNA packaging capacity, difficulty of vector production
Nonviral	Low immunogenic response, high loading capacity, chemical design flexibility, safe and stable, high transfection efficiency, biocompatible, easier to synthesize	Increased cytotoxicity for cationic lipids short half-lives, nonspecific binding to serum proteins for cationic carrier

**Table 2 pharmaceutics-14-00512-t002:** Some nonviral nanosystems for mRNA delivery in Section 5.

Category	Key Component	Core Technology	Size (nm)	Delivery Route	Gene	Ref
LNPs	Ionizable lipids, DOPE, C14-PEG2000	Heterocyclic lipids formed by one-step three-component reaction (3-CR)	100	In vitro i.m. (mouse)	Fluc	[75]
LNPs	DOPE, DSPC, PEG ionizable lipids	Helper lipid structure	170.5/167.2	i.v. (mouse)	Fluc	[78]
LNPs	C14–4, DOPE, Chol PEG	Optimized ratios in LNPs	57–151	In vitro	Luc	[81]
LNPs	Ionizable lipid, cholesterol, DSPC, DMPE-PEG	Optimized molar ratio between ionizable lipids and mRNA	82–90	In vitro	Human erythropoietin	[96]
LNPs	Synthesized lipid, DOPE, cholesterol, DMG-PEG2k	Lipid with unsaturated tail	143	In vitro i.v. (mouse)	Fluc	[97]
LNPs	DSPC, cholesterol, MC3, PEG2k-DMG	Generated via stepwise ethanol dilution	61	i.v. (rat) i.v. (monkey)	hEPO	[88]
LNPs	Cationic lipids, DSPE-PEG 2000	Optimized cationic lipid-like materials	140–160	DC-mediated (mouse)	OVA	[89]
Polymers	Ionizable lipids, phospholipids, PEG, cholesterol	Ionizable amphiphilic Janus dendrimer (IAJD)	75/92	In vitro i.p. (mouse)	Luc	[119]
Polymers	Alginate, chitosan	Hydrogels	Not mentioned	In vitro	hGLuc	[122]
Polymers	Chitosan, hyaluronic acid, trehalose	Hydrogels	80–180	In vitro	Luc	[124]
Polymers	PLGA, PEI	Nontoxic PLGA	428.9 ± 12	DC-based	GFP	[126]
Polymers	PEG	Cation-free delivery strategy with PEG	10–90	In vitro	GLuc	[127]
Polypeptide	Protamine	Natural cationic peptide	90–180	In vitro	Luc	[128]
Polypeptide	Protamine	Natural cationic peptide	30–110	i.d. (human)	TAA for NSCLC	[130]
Polypeptide	Pepfect14	Cationic CPP	70–110	i.p. (3D model)	eGFP mCherry	[134]
Polypeptide	RALA CPP	Arginine-rich peptide	89–144	In vitro i.v. (mouse)	eGFP OVA	[135]
Hybrid	Protamine, DOTAP	Heterogenous internal organization	146–234	In vitro; i.m. (mouse)	Luc	[141]
Hybrid	PLGA, DOTMA	Lipid-coated PLGA	231 ± 7.0	DC-based	m-cherry	[142]
Hybrid	PBAE, PEG-lipid	PBAE terpolymers formulated with PEG-lipid	200–370	i.v. (mouse)	Luc	[143]
Hybrid	Lipid, PLGA	adjuvant-loaded hybrid	300	DC-based i.v. (mouse)	EGFP	[144]
Hybrid	DP-7, DOTAP, cholesterol	DP7-C with double functions	100.23 ± 7.5	In vitro i.v./s.c. (mouse)	eGFP neoantigen	[145]
Polypeptide	PLA-NP, LAH4-L1	Cationic CPP	220.1 ± 22.9	In vitro	eGFP	[146]
Polypeptide	PEG, KL4 peptide	Monodisperse linear PEG with peptide	467.9 ± 24.9	In vitro i.n. (mouse)	eGFP	[147]
AuNP–DNA conjugates	AuNP, DNA	Targeted DNA	Not mentioned	In vitro s.c. (mouse)	BAX GFP	[159]
Exosome	Exosome	HEK-293F derived Exo	120 ± 50	i.m. (mouse)	Antares2	[167]
Exosome	Exosome, EVHB	HER2-targeted peptide	Not mentioned	i.p. (mouse)	HChrR6	[168]
Exosome	Exosomes	Exosome derived from RBC devoid of DNA	100–250	In vitro i.p., i.t. (mouse)	125b ASO	[173]

**Table 3 pharmaceutics-14-00512-t003:** Clinical trials of mRNA vaccine mediated by DC vaccines.

Company or Institution	Delivery Vehicle	Tumor Types	Antigens	Phase	Status	NCT Number
Oslo University Hospital	Not mentioned	Prostate cancer	-	Phase I/II	Completed	NCT01278914
Inge Marie Svane	Not mentioned	Breast cancer Malignant melanoma	Survivin, hTERT, p53	Phase I	Completed	NCT00978913
Oslo University Hospital	Not mentioned	Glioblastoma Brain tumor	Tumor stem cell	Phase I/II	Completed	NCT00846456
Oslo University Hospital	Not mentioned	Prostate cancer	hTERT, Survivin	Phase I/II	Active, not recruiting	NCT01197625
Radboud University	Electroporated	Uveal melanoma	Tyrosinase, gp100	Phase I/II	Terminated	NCT00929019
National Cancer Institute	Not mentioned	Leukemia	-	Phase I	Terminated	NCT00514189
Steinar Aamdal	Not mentioned	Recurrent epithelial ovarian cancer	hTERT, survivin	Phase I/II	Terminated	NCT01334047
Steinar Aamdal	Not mentioned	Metastatic malignant melanoma	hTERT, survivin	Phase I/II	Terminated	NCT00961844
Trinomab Biotech Co., Ltd.	Not mentioned	Brain cancer, neoplasm metastases	-	Phase I	Unknown	NCT02808416
University of Florida	Not mentioned	Metastatic prostate cancer	hTERT	Phase I/II	Withdrawn	NCT01153113
Inge Marie Svane	Electroporated	Prostatic neoplasms	PSA, PAP, survivin, hTERT	Phase II	Completed	NCT01446731
Radboud University	Electroporated	Colorectal cancer, liver Metastases	Carcinoembryonic antigen	Phase I/II	Completed	NCT00228189
National Cancer Institute	Not mentioned	Recurrent central nervous, system neoplasm	Brain tumor stem cell-specific mRNA	Phase I	Completed	NCT00890032
University Hospital, Antwerp	Not mentioned	Glioblastoma, renal cell carcinoma, sarcomas, breast cancers, malignant mesothelioma, colorectal tumors	WT1 protein	Phase I/II	Unknown	NCT01291420
Memorial Sloan Kettering Cancer Center	Electroporated	Melanoma	Tumor-associated antigen	Phase I	Active, not recruiting	NCT01456104
Affiliated Hospital to Academy of Military Medical Sciences	Not mentioned	Esophagus cancer	MUC1, survivin	Phase I/II	Unknown	NCT02693236
National Cancer Institute	Not mentioned	Malignant neoplasms of brain	pp65-LAMP	Phase I	Active, not recruiting	NCT00639639
University Hospital, Antwerp	Electroporated	Acute myeloid leukemia	Wilms’ tumor antigen 1	Phase I	Completed	NCT00834002
Life Research Technologies GmbH	Not mentioned	Ovarian epithelial cancer	TERT-	Phase I	Unknown	NCT01456065
Memorial Sloan Kettering Cancer Center	Electroporated	Multiple myeloma	CT7, MAGE-A3, WT1	Phase I	Active, not recruiting	NCT01995708
Radboud University	Not mentioned	Melanoma stage III or IV	gp100 and tyrosinase	Phase I/II	Completed	NCT00243529
Radboud University	Electroporated	Hematological malignancies	Minor histocompatibility antigens	Phase I/II	Completed	NCT02528682
National Cancer Institute	Not mentioned	Malignant neoplasms brain	Cytomegalovirus (CMV) pp65-lysosome-associated membrane protein (LAMP)	Phase I	Completed	NCT00626483
Zwi Berneman	Electroporated	Acute myeloid leukemia	Wilms’ tumor antigen (WT1)	Phase II	Recruiting	NCT01686334
CureVac AG	Not mentioned	Non-small-cell lung cancer	-	Phase I/II	Completed	NCT00923312
Radboud University	Protamine	Prostatic neoplasms	-	Phase II	Completed	NCT02692976
Oslo University Hospital	Not mentioned	Glioblastoma	Survivin and hTERT	Phase II/III	Recruiting	NCT03548571
Ludwig Maximilian University of Munich	Electroporated	Acute myeloid leukemia	WT1, PRAME, CMVpp65	Phase I/II	Completed	NCT01734304
University Hospital, Antwerp	Electroporated	Malignant pleural mesothelioma	Wilms’ tumor protein 1 (WT1)	Phase I/II	Recruiting	NCT02649829
Asterias Biotherapeutics, Inc.	Not mentioned	Acute myelogenous leukemia	hTERT and a portion of the lysosome-associated membrane protein (LAMP-1)	Phase II	Completed	NCT00510133
Oslo University Hospital	Not mentioned	Malignant melanoma	-	Phase I/II	Completed	NCT01278940
Radboud University	Electroporated	Melanoma	gp100 and tyrosinase	Phase I/II	Completed	NCT01530698
University Hospital, Antwerp	Electroporated	High-grade glioma, diffuse intrinsic pontine glioma	WT1	Phase I/II	Recruiting	NCT04911621
Radboud University	Electroporated	Melanoma	gp100 and tyrosinase	Phase I/II	Completed	NCT00940004
Bart Neyns	Electroporated	Melanoma	-	Phase I	Completed	NCT01066390
Radboud University	Electroporated	Melanoma	gp100 and tyrosinase	Phase II	Completed	NCT02285413
Immunomic Therapeutics, Inc.	Electroporated	Glioblastoma multiforme, glioblastoma, malignant glioma, astrocytoma, grade IVGBM	pp65-shLAMP	Phase II	Recruiting	NCT02465268
University Hospital, Antwerp	Electroporated	Glioblastoma multiforme of brain	WT1	Phase I/II	Recruiting	NCT02649582
Gary Archer, Ph.D.	Not mentioned	Glioblastoma	Human CMV pp65-LAMP	Phase II	Recruiting	NCT03688178
University Hospital, Antwerp	Electroporated	Acute myeloid leukemia	WT1	Phase I	Completed	NCT00834002
Gary Archer, Ph.D.	Electroporated	Glioblastoma	Human CMV pp65-LAMP	Phase II	Suspended	NCT03927222
Universitair Ziekenhuis Brussel	Electroporated	Malignant melanoma	-	Phase II	Completed	NCT01676779
Guangdong 999 Brain Hospital	Not mentioned	Glioblastoma	Personalized TAAs	Phase I	Active, Not recruiting	NCT02808364

**Table 4 pharmaceutics-14-00512-t004:** The clinical trials of mRNA cancer vaccine based on direct injection.

Company or Institution	Delivery Vehicle	Tumor Types	Antigens	Phase	Status	NCT Number
CureVac AG	Protamine	Non-small-cell lung cancer	MUC1, survivin, NY-ESO-1, 5T4, MAGE-C2, MAGE-C1	Phase I/II	Completed	NCT03164772
University Hospital Tuebingen	Not mentioned	Malignant melanoma	Melan-A, Mage-A1, Mage-A3, Survivin, GP100, tyrosinase	Phase I/II	Completed	NCT00204516
University Hospital Tuebingen	Protamine	Malignant melanoma	Melan-A, Mage-A1, Mage-A3, Survivin, GP100, and tyrosinase	Phase I/II	Completed	NCT00204607
Merck	Not mentioned	Non-small-cell lung cancer, pancreatic neoplasms, colorectal neoplasms	mRNA-5671/V941	Phase I	Recruiting	NCT03948763
BioNTech	Lipid-based vector	Ovarian cancer	Ova	Phase I	Recruiting	NCT04163094
Moderna	Lipid-based vector	Melanoma	Personalized mRNA-4157	Phase II	Recruiting	NCT03897881
Moderna	Lipid-based vector	Solid tumors	Personalized mRNA-4157	Phase I	Recruiting	NCT03313778
BioNTech	Lipid-based vector	Prostate cancer	W_pro1	Phase I/II	Recruiting	NCT04382898
Universitair Ziekenhuis Brussel	Naked mRNA	Early-stage breast cancer	-	Phase I	Recruiting	NCT03788083
National Cancer Institute	Lipid-based vector	Melanoma, colon cancer, gastrointestinal cancer, genitourinary cancer, hepatocellular cancer	Personalized mRNA	Phase I/II	Terminated	NCT03480152
University Hospital Tuebingen	Protamine	Recurrent prostate cancer	-	Phase I/II	Unknown	NCT02452307
University of Florida	Lipid-based vector	Adult glioblastoma	Autologous total tumor and pp65 full-length (fl) lysosome-associated membrane protein (LAMP)	Phase I	Recruiting	NCT04573140
Moderna	Lipid-based vector	Relapsed/refractory solid tumor malignancies or lymphoma	human OX40L	Phase I/II	Active, not recruiting	NCT03323398
BioNTech	Lipid-based vector	Melanoma	NY-ESO-1, tyrosinase, MAGE-A3, TPTE	Phase I	Active, not recruiting	NCT02410733
BioNTech SE	Lipid-based vector	Breast cancer	Tumor relevant and immunogenic RNA	Phase I	Active, not recruiting	NCT02316457
CureVac AG	Peptide-based vector	Non-small-cell lung cancer	CV9201	Phase I/II	Completed	NCT00923312
CureVac AG	Peptide-based vector	Non-small-cell lung cancer	CV9202	Phase I	Terminated	NCT01915524
CureVac AG	Peptide-based vector	Prostate cancer	CV9103	Phase I/II	Completed	NCT00831467
CureVac AG	Peptide-based vector	Prostate cancer	CV9103	Phase I/II	Terminated	NCT00906243
CureVac AG	Peptide-based vector	Prostate cancer	CV9104	Phase I/II	Terminated	NCT01817738
CureVac AG	Peptide-based vector	Prostate cancer	CV9104	Phase II	Terminated	NCT02140138
Moderna	Lipid-based vector	Solid tumor malignancies or lymphoma	mRNA-2752	Phase I	Recruiting	NCT03739931
Stemirna Therapeutics	Not mentioned	Esophageal cancer, non-small-cell lung cancer	Personalized mRNA	Not applicable	Not yet recruiting	NCT03908671
Changhai Hospital	Not mentioned	Advanced Esophageal Squamous; Carcinoma; Gastric Adenocarcinoma, pancreatic adenocarcinoma, colorectal adenocarcinoma	Personalized mRNA	Not applicable	Recruiting	NCT03468244
Laura Esserman	Not mentioned	Carcinoma, intraductal, noninfiltrating	mRNA 2752	Phase I	Recruiting	NCT02872025
University Medical Center Groningen	Not mentioned	Cervical cancer	HPV-derived tumor antigens	Phase I	Completed	NCT03141463

## Data Availability

Not applicable.

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
