# Peer review of "Nonviral Delivery Systems of mRNA Vaccines for Cancer Gene Therapy"

_pharmaceutics, 2022, doi:10.3390/pharmaceutics14030512_

Round 1

Reviewer 1 Report

The work entitled “Nonviral Delivery Systems of mRNA Vaccines for Cancer Gene Therapy” by Wang et al. focuses on nonviral nanodelivery systems of mRNA vaccines used for cancer gene therapy and immunotherapy. Over the years many mechanisms and processes for delivering of mRNA vaccines have been uncovered; however, this is an area that continues to grow and expand, with new strategies appearing each day. This review highlights that expanding field of research and the emerging novelties.

The work is very well organized and the research aside from being pertinent, to the point and up to date is also scientifically sound. The authors did an excellent job getting deeper and deeper in their investigation. It would have been however, extremely important to have in the introduction section a sentence or a paragraph in which the author highlighted the novelty of this research and its importance compared to the many reviews already in existence, also in this field. Also, there are instances when we clearly see information being presented in sequential mode instead of a more organic way, meaning that there are instances when the authors present the information given by one author, than the other, and the other but there is little connection and that would be most important to improve the quality of the manuscript. There are also English writing mistakes that should be fixed; however, none too important to compromise the meaning and scientific quality of the information. Finally, even though the authors made a significant effort in presenting many future ideas and predictions for the field, section 7 is way too extensive, leaving the readers tired and without reaching an ideal climax after reading such a well though out manuscript. This section requires more refinement and attention and should be abbreviated.

Overall, the work is of high quality and should be considered for publication after minor revision.

Author Response

Dear  Reviewer:

Thank you for your letter and for the reviewer’s comments concerning our manuscript. Those comments are all valuable and very helpful for revising and improving our manuscript. We have studied comments carefully and have made correction which we hope meet with approval. The main corrections in the paper and responds to the reviewer’s comments are as following (We also uploaded an attachment):

Point 1. It would have been however, extremely important to have in the introduction section a sentence or a paragraph in which the author highlighted the novelty of this research and its importance compared to the many reviews already in existence, also in this field.

Response 1: Thank you for reading our manuscript carefully. Some sentences which highlighted the novelty of our research were added into paper and the addition begins with line 160 of the revised manuscript. The contents are as follows: On the other hand, in this review, we systematically compiled the nonviral nano-delivery systems currently used in mRNA vaccines with analysis of their advantages and disadvantages, which may help future mRNA vaccine development on vector selecting. In addition, we also comprehensively listed the mRNA cancer vaccines in various clinical trials, providing some updated information.

Point 2. Also, there are instances when we clearly see information being presented in sequential mode instead of a more organic way, meaning that there are instances when the authors present the information given by one author, than the other, and the other but there is little connection and that would be most important to improve the quality of the manuscript.

Response: Thank you for reading our manuscript carefully and your suggestion. We have tried to present the examples cited by our manuscript in a more organic way. For instance, in the section 5.1. Lipid nanoparticles (LNPs), we put examples together, which have the same stated purpose, and we add linking words/phrases between them, as shown below: Despite some insufficiencies, mRNA-loaded NPs are still potential clinical tools, as demonstrated by unprecedented rapid development of mRNA COVID-19 vaccines. To make LNPs more powerful in mRNA delivery, many scientists tried to optimize them, and there have been some new findings. For transfection efficiency, Kauffman et al. showed that increasing the ionizable lipid: mRNA weight ratio can enhance delivery efficiency [91]. Ball et al. optimized the LNP delivery system by adding a negatively charged “helper molecular” to the NP. In their study, the mRNA-loaded LNPs co-formulated with siRNA induced three-fold increases in luciferase protein expression compared to that in formulations without siRNA [92]. For endosomal escape, Herrera M. et al. found that it preferentially occurs in late endosomes, not early endosomes [93]. Maugeri M. et al. demonstrated that endosomal escape of mRNA-loaded LNPs depends on the molar ratio between the ionizable lipids and mRNA nucleotides [94]. Lee, S. M et al. studied the interaction between LNPs and a model endosomal membrane and showed that 4A3-Cit (a lipid with an unsaturated tail) exhibited superior lipid fusion over saturated lipids, suggesting that unsaturated lipids promote endosomal escape [95]. For targeted delivery, scientists have observed that the targeting functionalities of LNPs are largely related to the chemical structure of the active lipids [96]. For instance, imidazole-based LNPs preferentially target splenic T cells [97]; Zukancic et al. found that PEGylation is critical for achieving selective organ targeting, even though it is at the lowest ratio in LNPs [98]. For immunogenicity, Hassett et al. found that the particle size may influence LNP immunogenicity; they demonstrated that LNPs in smaller diameter were substantially less immunogenic in mice, but all the particle sizes tested induced a robust immune response in nonhuman primates (NHPs), suggesting that an optimal mRNA vaccine particle size as determined for rodents may not translate to primates [99]. For toxicity, previous research has shown that nonbiodegradable lipids would cause mortality in mice, whereas biodegradable and biodegradable lipids administered at a similar dose were well tolerated [100].

 Point 3. There are also English writing mistakes that should be fixed.

Response: Thank you for reading our manuscript carefully and your suggestion. We have corrected all spelling errors in the manuscript (including tumour, titres, programme). We also corrected some English writing mistakes. For example, we changed “Therefore IVT mRNAs need to be designed without GC-enriched 5'-UTRs.” to “Therefore, IVT mRNAs need to be designed without GC-enriched 5'-UTRs.”. Moreover, we corrected some wrong tenses in the paper. For example, we changed “Lipid-polymer hybrid nanoparticles (LPNs) were the earliest reported example of hybrid NPs” to “Lipid-polymer hybrid nanoparticles (LPNs) are the earliest reported example of hybrid NPs”. In this revision, we have re-edited the language of the manuscript and sent it to AJE, a professional organization, for language polishing and modification. The language modification certificate is attached. The picture is shown in the attached document.

Point 4. Finally, even though the authors made a significant effort in presenting many future ideas and predictions for the field, section 7 is way too extensive, leaving the readers tired and without reaching an ideal climax after reading such a well though out manuscript. This section requires more refinement and attention and should be abbreviated.

Response: Thank you for reading our manuscript carefully and your suggestion. We shortened the section 7 by removing some contents similar to those appeared in the introduction, with the number of words reduced from 1528 to 967. In addition, we also rearranged the section 7 to present the information in a more organic way. In section 7, we followed an order of current situation, drawbacks and future direction of the mRNA cancer vaccines.

All in all, thank you very much for your valuable suggestions. We tried our best to improve the manuscript and made some changes in the manuscript. And we appreciate for your warm work earnestly, and hope that the correction will meet with approval.

Sincerely yours,

Prof Yang Li

State Key Laboratory of Biotherapy

Sichuan University, Chengdu, P. R. China

Reviewer 2 Report

The manuscript "Nonviral Delivery Systems of mRNA Vaccines for Cancer Gene Therapy" is a review paper that reports the use of messenger RNA for cancer therapy. This is a hot topic subject in the field of nanomedicine and of great interest to the readers. I consider the manuscript is well documented and can be considered for publication in Pharmaceutics, after the authors reply. Some comments are given hereunder:

  • expressions like "in vivo", "ex vivo", "in vitro" should be written in italic;
  • I would suggest to cite the reference: Pharmaceutics 2021,13, 2090. https://doi.org/10.3390/pharmaceutics1312209
  • where are documented the information given in Table3. The clinical trials of mRNA vaccine mediated by DC vaccines. Their origin should be given in the manuscript.
  • the same comment is available for information given in Table 4.
  • the figures 2 and 4 are from reference or the authors drawn them?

Author Response

Dear Reviewer:

Thank you for your letter and for the reviewer’s comments concerning our manuscript. Those comments are all valuable and very helpful for revising and improving our manuscript. We have studied comments carefully and have made correction which we hope meet with approval. The main corrections in the paper and responds to the reviewer’s comments are as following (We also uploaded an attachment):

Point 1: expressions like "in vivo", "ex vivo", "in vitro" should be written in italic.

Response1: Thank you for reading our manuscript carefully. All expression like "in vivo", "in vitro", "ex vivo" in the paper have been written in italic.

Point 2: I would suggest to cite the reference: Pharmaceutics 2021,13, 2090. https://doi.org/10.3390/pharmaceutics13122090

Response 2: Thank you for reading our manuscript carefully and your suggestion. We have cited this reference in the section 4 (The principles of mRNA vaccine design and modification) of our manuscript. The contents of the reference section are shown as follows: Discovered in the 1970s, the 5'-cap structure (m7G5'ppp5'N), composed of a 7-methylguanosine nucleoside and a terminal nucleotide linked through a triphosphate bridge in the 5'-mRNA, confers IVT mRNA stability and translation efficiency [1]; and the 5'-UTR is a noncoding region, but it can help mRNA bind to ribosomes [1].  

Point 3. where are documented the information given in Table3. The clinical trials of mRNA vaccine mediated by DC vaccines. Their origin should be given in the manuscript. the same comment is available for information given in Table 4.

Response3: Thank you for reading our manuscript carefully and your suggestion. All the information given in Table 3 and 4 are found in https://clinicaltrials.gov, and we also added this in the section 6 of our manuscript, just as follows: In this section, we most focus on the clinical and preclinical trials of these mRNA cancer vaccines, which can be found on the website of https://clinicaltrials.gov.

Point 4: the figures 2 and 4 are from reference or the authors drawn them?

Response4: Thank you for reading our manuscript carefully. The figure 2 was drawn by ourselves by referring to several references, and the figure 4 is from a reference [2] (Liu T, Liang Y, Huang L. Development and Delivery Systems of mRNA Vaccines. Front Bioeng Biotechnol. 2021;9:718753. Published 2021 Jul 27. doi:10.3389/fbioe.2021.718753). We also added and explained it in the annotation of figure 4, shown as bellow: Figure 4. The structure of the 5'-cap (cited from the reference [43]). The first nucleoside in the 5'-cap is usually composed of a guanine methylated at the seventh position and a ribosome (m7G) ……

All in all, thank you very much for your valuable suggestions. We tried our best to improve the manuscript and made some changes in the manuscript. And we appreciate for your warm work earnestly, and hope that the correction will meet with approval.

Sincerely yours,

Prof Yang Li

State Key Laboratory of Biotherapy

Sichuan University, Chengdu, P. R. China

Reference:

  1. Baptista, B.; Carapito, R.; Laroui, N.; Pichon, C.; Sousa, F. mRNA, a Revolution in Biomedicine. Pharmaceutics 2021, 13, doi:10.3390/pharmaceutics13122090.
  2. Liu, T.; Liang, Y.; Huang, L. Development and Delivery Systems of mRNA Vaccines. Front Bioeng Biotechnol 2021, 9, 718753, doi:10.3389/fbioe.2021.718753.

Reviewer 3 Report

The review paper entitled: “Nonviral Delivery Systems of mRNA Vaccines for Cancer Gene Therapy”, is overall interesting as it deals with a topic of high clinical impact.

However, although the authors try to be exhaustive in the descriptions of the contents in the various paragraphs, the data reported in this manuscript do not provide an added value compared to those published and still available in literature, see for example https://pubmed.ncbi.nlm.nih.gov/32013049/ and https://pubmed.ncbi.nlm.nih.gov/33632261/ which have not been cited by the authors.

In order to make the review more relevant and also more updated on the main topic, i.e. use of mRNA vaccines for cancer gene therapy, the authors should improve and implement the Section 6 (Clinical mRNA vaccines for cancer therapy) to highlight the novelty of the manuscript. On the other hand, the authors should also shorten the previous sections taking into account what has already been published on the same topic.  

In addition, the tumor associated antigens codified by specific mRNA vaccines used and cited by clinical trials should be specified (in the Table 3 and 4 and explained in the text) when these information are available in literature.

Author Response

Dear Reviewer:

Thank you for your letter and for the reviewer’s comments concerning our manuscript. Those comments are all valuable and very helpful for revising and improving our manuscript. We have studied comments carefully and have made correction which we hope meet with approval. The main corrections in the paper and responds to the reviewer’s comments are as following (We also upload an attachment):

Point 1: the authors should improve and implement the Section 6 (Clinical mRNA vaccines for cancer therapy) to highlight the novelty of the manuscript.

Response 1: Thank you for reading our manuscript carefully and your suggestion. To make the section 6 more complete and logical, we added some contents about the employed vectors of both DCs-based mRNA cancer vaccines and direct-injection-based mRNA cancer vaccines. In addition, we tried to add and analyse more clinical trials which have been presented in the Table 3/4 in the manuscript, including the completed, active and terminated ones.

Point 2. the authors should also shorten the previous sections taking into account what has already been published on the same topic.

Response 2: Thank you for reading our manuscript carefully and your suggestion. Taking into account the information which has already been published, we have shortened the sections from 1 to 5 by removing some contents. The number of words in the first five sections has been reduced from 8714 to nearly 6700.

Point 3. In addition, the tumor associated antigens codified by specific mRNA vaccines used and cited by clinical trials should be specified (in the Table 3 and 4 and explained in the text) when these information are available in literature.

Response 3: Thank you for reading our manuscript carefully and your suggestion. We have added the tumor associated/specific antigens codified by mRNA vaccines cited by clinical trials both in the Table 3/4 and the manuscript. In addition, some specific information about clinical trials is also added.

Table 3. The clinical trials of mRNA vaccine mediated by DC vaccines.

Company or institution

Delivery vehicle

Tumor types

Antigens

Phase

Status

NCT number

Oslo University Hospital

Not mentioned

Prostate Cancer

/

Phase I/II

Completed

NCT01278914

Inge Marie Svane

Not mentioned

Breast Cancer

Malignant Melanoma

Survivin, Htert, p53

Phase I

Completed

NCT00978913

Oslo University Hospital

Not mentioned

Glioblastoma

Brain Tumor

Tumor Stem Cell

Phase I/II

Completed

NCT00846456

Oslo University Hospital

Not mentioned

Prostate Cancer

hTERT, Survivin

Phase I/II

Active,

not recruiting

NCT01197625

Radboud University

Electroporated

Uveal Melanoma

tyrosinase,gp100

Phase I/II

Terminated

NCT00929019

National Cancer Institute

Not mentioned

Leukemia

/

Phase I

Terminated

NCT00514189

Steinar Aamdal

Not mentioned

Recurrent Epithelial Ovarian Cancer

hTERT, Survivin

Phase I/II

Terminated

NCT01334047

Steinar Aamdal

Not mentioned

Metastatic Malignant Melanoma

hTERT, Survivin

Phase I/II

Terminated

NCT00961844

Trinomab Biotech Co, Ltd.

Not mentioned

Brain Cancer; Neoplasm Metastases

/

Phase I

Unknown

NCT02808416

University of Florida

Not mentioned

Metastatic Prostate Cancer

hTERT

Phase I/II

Withdrawn

NCT01153113

Inge Marie Svane

Electroporated

Prostatic Neoplasms

PSA, PAP, survivin and hTERT

Phase II

Completed

NCT01446731

Radboud University

Electroporated

Colorectal Cancer; Liver Metastases

carcinoembryonic antigen

Phase I/II

Completed

NCT00228189

National Cancer Institute

Not mentioned

Recurrent Central Nervous;

System Neoplasm

brain tumor stem cells specific mRNA

Phase I

Completed

NCT00890032

University Hospital, Antwerp

Not mentioned

Glioblastoma; Renal Cell Carcinoma; Sarcomas; Breast Cancers; Malignant Mesothelioma; Colorectal Tumors

WT1 protein

Phase I/II

Unknown

NCT01291420

Memorial Sloan Kettering Cancer Center

Electroporated

Melanoma

Tumor-associated Antigen 

Phase I

Active,

not recruiting

NCT01456104

Affiliated Hospital to Academy of Military Medical Sciences

Not mentioned

Esophagus Cancer

MUC1 and Survivin

Phase I/II

Unknown

NCT02693236

National Cancer Institute

Not mentioned

Malignant Neoplasms of Brain

pp65-LAMP

Phase I

Active,

not recruiting

NCT00639639

University Hospital, Antwerp

Electroporated

Acute Myeloid Leukemia

Wilms tumor antigen-1

Phase I

Completed

NCT00834002

Life Research Technologies GmbH

Not mentioned

Ovarian Epithelial Cancer

TERT-

Phase I

Unknown

NCT01456065

Memorial Sloan Kettering Cancer Center

Electroporated

Multiple Myeloma

CT7, MAGE-A3, and WT1 

Phase I

Active,

not recruiting

NCT01995708

Radboud University

Not mentioned

Melanoma Stage III or IV

gp100 and tyrosinase

Phase I/II

Completed

NCT00243529

Radboud University

Electroporated

Hematological Malignancies

minor histocompatibility antigens

Phase I/II

Completed

NCT02528682

National Cancer Institute

Not mentioned

Malignant Neoplasms Brain

cytomegalovirus (CMV) pp65-lysosomal-associated membrane protein (LAMP) 

Phase I

Completed

NCT00626483

Zwi Berneman

Electroporated

Acute Myeloid Leukemia

Wilms' tumor antigen (WT1)

Phase II

Recruiting

NCT01686334

CureVac AG

Not mentioned

Non Small Cell Lung Cancer

/

Phase I/II

Completed

NCT00923312

Radboud University

Protamine

Prostatic Neoplasms

/

Phase II

Completed

NCT02692976

Oslo University Hospital

Not mentioned

Glioblastoma

 survivin and hTERT

Phase II/III

Recruiting

NCT03548571

Ludwig-Maximilians-University of Munich

Electroporated

Acute Myeloid Leukemia

WT1, PRAME, and CMVpp65

Phase I/II

Completed

NCT01734304

University Hospital, Antwerp

Electroporated

Malignant Pleural Mesothelioma

Wilms' Tumor Protein 1 (WT1)

Phase I/II

Recruiting

NCT02649829

Asterias Biotherapeutics, Inc.

Not mentioned

Acute Myelogenous Leukemia

hTERT and a portion of the lysosome-associated membrane protein LAMP-1 (LAMP)

Phase II

Completed

NCT00510133

Oslo University Hospital

Not mentioned

Malignant Melanoma

/

Phase I/II

Completed

NCT01278940

Radboud University

Electroporated

Melanoma

gp100 and tyrosinase

Phase I/II

Completed

NCT01530698

University Hospital, Antwerp

Electroporated

High Grade GliomaDiffuse Intrinsic Pontine Glioma

WT1

Phase I/II

Recruiting

NCT04911621

Radboud University

Electroporated

Melanoma

gp100 and tyrosinase

Phase I/II

Completed

NCT00940004

Bart Neyns

Electroporated

Melanoma

/

Phase I

Completed

NCT01066390

Radboud University

Electroporated

Melanoma

gp100 and tyrosinase

Phase II

Completed

NCT02285413

Immunomic Therapeutics, Inc.

Electroporated

Glioblastoma Multiforme; Glioblastoma; Malignant Glioma; Astrocytoma, Grade IVGBM

pp65-shLAMP 

Phase II

 Recruiting

NCT02465268

University Hospital, Antwerp

Electroporated

Glioblastoma Multiforme of Brain

WT1

Phase I/II

Recruiting

NCT02649582

Gary Archer Ph.D

Not mentioned

Glioblastoma

human CMV pp65-LAMP 

Phase II

Recruiting

NCT03688178

University Hospital, Antwerp

Electroporated

Acute Myeloid Leukemia

WT1

Phase I

Completed

NCT00834002

Gary Archer Ph.D.

Electroporated

Glioblastoma

Human CMV pp65-LAMP 

Phase II

Suspended

NCT03927222

Universitair Ziekenhuis Brussel

Electroporated

Malignant Melanoma

/

Phase II

Completed

NCT01676779

Guangdong 999 Brain Hospital

Not mentioned

Glioblastoma

Personalized TAAs

Phase I

Active,

Not recruiting

NCT02808364

Table 4. The clinical trials of mRNA cancer vaccine based on direct injection.

Company or

institution

Delivery vehicle

Tumor types                                 

Antigens

Phase

Status

NCT number

CureVac AG

Protamine

Non-Small-Cell Lung cancer

MUC1, survivin, NY-ESO-1, 5T4, MAGE-C2 and MAGE-C1

Phase I/II

Completed

NCT03164772

University Hospital Tuebingen

Not mentioned

Malignant Melanoma

Melan-A, Mage-A1, Mage-A3, Survivin, GP100 and Tyrosinase

Phase I/II

Completed

NCT00204516

University Hospital Tuebingen

Protamine

Malignant Melanoma

Melan-A, Mage-A1, Mage-A3, Survivin, GP100 and Tyrosinase

Phase I/II

Completed

NCT00204607

Merck

Not mentioned

Non-Small-Cell Lung cancer

Pancreatic Neoplasms

Colorectal Neoplasms

mRNA-5671/V941

Phase I

Recruiting

NCT03948763

BioNTech

Lipid-based vector

Ovarian Cancer

Ova

Phase I

Recruiting

NCT04163094

Moderna

Lipid-based vector

Melanoma

 Personalized mRNA-4157

Phase II

Recruiting

NCT03897881

Moderna

Lipid-based vector

Solid Tumors

 Personalized mRNA-4157

Phase I

Recruiting

NCT03313778

BioNTech

Lipid-based vector

Prostate Cancer

W_pro1

Phase I/II

Recruiting

NCT04382898

Universitair Ziekenhuis Brussel

Naked mRNA

Early-stage Breast Cancer

/

Phase I

Recruiting

NCT03788083

National Cancer

Institute 

Lipid-based vector

Melanoma; Colon Cancer; Gastrointestinal Cancer; Genitourinary Cancer; Hepatocellular Cancer

Personalized  mRNA

Phase I/II

Terminated

NCT03480152

University Hospital Tuebingen

Protamine

Recurrent Prostate Cancer

/

Phase I/II

Unknown

NCT02452307

University of Florida

Lipid-based vector

Adult Glioblastoma

Autologous total tumor and pp65 full length (fl) lysosomal associated membrane protein (LAMP)

Phase I

Recruiting

NCT04573140

Moderna

Lipid-based vector

Relapsed/refractory solid tumor malignancies or lymphoma

human OX40L

Phase I/II

Active, not recruiting

NCT03323398

BioNTech

Lipid-based vector

Melanoma

NY-ESO-1, Tyrosinase,

MAGE-A3, and TPTE

Phase I

Active, not recruiting

NCT02410733

BioNTech SE

Lipid-based vector

Breast Cancer

Tumor relevant and immunogenic RNA

Phase I

Active, not recruiting

NCT02316457

CureVac AG

Peptide-based vector

Non Small Cell Lung Cancer

CV9201

Phase I/II

Completed

NCT00923312

CureVac AG

Peptide-based vector

Non Small Cell Lung Cancer

CV9202

Phase I

Terminated

NCT01915524

CureVac AG

Peptide-based vector

Prostate cancer

CV9103

Phase I/II

Completed

NCT00831467

CureVac AG

Peptide-based vector

Prostate cancer

CV9103

Phase I/II

Terminated

NCT00906243

CureVac AG

Peptide-based vector

Prostate cancer

CV9104

Phase I/II

Terminated

NCT01817738

CureVac AG

Peptide-based vector

Prostate cancer

CV9104

Phase II

Terminated

NCT02140138

Moderna

Lipid-based vector

Solid Tumor Malignancies or Lymphoma

mRNA-2752

Phase I

Recruiting

NCT03739931

Stemirna Therapeutics

Not mentioned

Esophageal CancerNon Small Cell Lung Cancer

Personalized mRNA

Not Applicable

Not yet recruiting

NCT03908671

Changhai Hospital

Not mentioned

Advanced Esophageal Squamous; Carcinoma; Gastric Adenocarcinoma; Pancreatic Adenocarcinoma; Colorectal Adenocarcinoma

Personalized mRNA

Not Applicable

Recruiting

NCT03468244

Laura Esserman

Not mentioned

Carcinoma, Intraductal;

Noninfiltrating

mRNA 2752

Phase I

Recruiting

NCT02872025

University Medical Center Groningen

Not mentioned

Cervical Cancer

HPV-derived tumor

antigens

Phase I

completed

NCT03141463

Additions in the manuscript:

  • Moreover, Wang Q. T. et al. developed a DC vaccine against personalized TAAs to treat patients with glioblastoma multiforme (GBM) or advanced lung cancer in combination with low-dose cyclophosphamide, polyinosinic-polycytidylic acid (poly I:C), imiquimod and an anti-PD-1 antibody (NCT02808364).
  • For the melanoma FixVac (BNT111), an intravenously administered liposomal RNA (encoding antigens: NY-ESO-1, Tyrosinase, MAGE-A3, and TPTE) vaccine developed by BioNTech, the data from an exploratory interim analysis showed that melanoma FixVac, alone or in combination with blockade of the checkpoint inhibitor PD1, mediated durable objective responses (NCT02410733)
  • Moreover, in a phase I trial of Vvax001 (NCT03141463), an saRNA encoding HPV-derived tumor antigens cancer vaccine against human papillomavirus (HPV)-induced cancers, immunological activity, safety, and tolerability were detected.

All in all, thank you very much for your valuable suggestions. We tried our best to improve the manuscript and made some changes in the manuscript. And we appreciate for your warm work earnestly, and hope that the correction will meet with approval.

Sincerely yours,

Prof Yang Li

State Key Laboratory of Biotherapy

Sichuan University, Chengdu, P. R. China

Round 2

Reviewer 3 Report

The authors have fully addressed all changes and suggestions required to improve the manuscript. In my opinion, the revised manuscript is now suitable for publication in present form.